# STYLEAR: CUSTOMIZING MULTIMODAL AUTORE-GRESSIVE MODEL FOR STYLE-ALIGNED TEXT-TO-IMAGE GENERATION

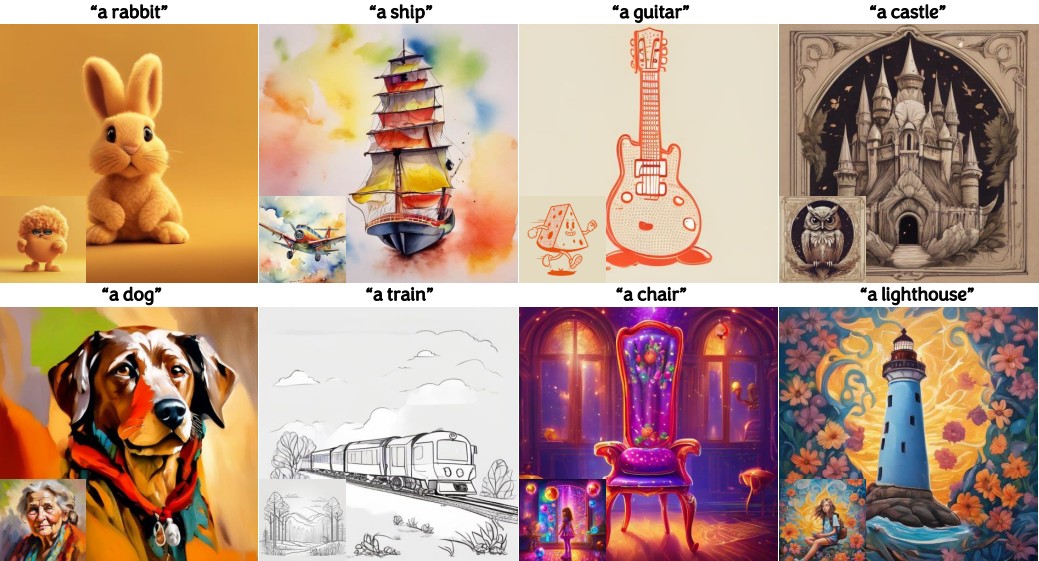

Figure 1: **Stylized samples of our StyleAR.** Our StyleAR is capable of generating images that are highly consistent in style with the reference images across a diverse range of styles, and highly aligned in semantics with the input prompts of various categories.

## ABSTRACT

In the current research landscape, multimodal autoregressive (AR) models have shown exceptional capabilities across various domains, including visual understanding and generation. However, complex tasks such as style-aligned text-to-image generation present significant challenges, particularly in data acquisition. In analogy to instruction-following tuning for image editing of AR models, style-aligned generation requires a reference style image and prompt, resulting in a text-image-to-image triplet where the output shares the style and semantics of the input. However, acquiring large volumes of such triplet data with specific styles is considerably more challenging than obtaining conventional text-to-image data used for training generative models. To address this issue, we propose StyleAR, an innovative approach that combines a specially designed data curation method with our proposed AR models to effectively utilize text-to-image binary data for style-aligned text-to-image generation. Our method synthesizes target stylized data using a reference style image and prompt, but only incorporates the target stylized image as the image modality to create high-quality binary data. To facilitate binary data training, we introduce a CLIP image encoder with a perceiver resampler that translates the image input into style tokens aligned with multimodal tokens in AR models and implement a style-enhanced token technique to prevent content leakage which is a common issue in previous work. Furthermore, we mix raw images drawn from large-scale text-image datasets with stylized images

to enhance StyleAR's ability to extract richer stylistic features and ensure style consistency. Extensive qualitative and quantitative experiments demonstrate our superior performance.

# 1 INTRODUCTION

Multimodal autoregressive (AR) models (Team, 2024; Liu et al., 2024; Chern et al., 2024; Wang et al., 2024b; Chen et al., 2025; Sun et al., 2024) have revolutionized cross-modal content synthesis by using the next-token prediction mechanism, demonstrating superior performance in tasks such as image understanding and text-to-image generation. In particular, in text-to-image generation, AR models outperform traditional diffusion models (Rombach et al., 2022; Podell et al., 2023; Esser et al., 2024; Labs, 2024; Xiao et al., 2024; Xie et al., 2024; Zhou et al., 2024; Wu et al., 2024) in terms of prompt adherence and image quality. However, when tackling style-aligned text-to-image generation, a task requiring the style and semantic of generated images precise alignment with reference style images and input prompts, respectively, AR models exhibit significant performance gaps compared to diffusion-based approaches (Hertz et al., 2024; Wang et al., 2024a; Ye et al., 2023; Jeong et al., 2024; Sohn et al., 2023; Huang et al., 2025; Liu et al., 2023; Wang et al., 2023; Ruiz et al., 2023; Gao et al., 2024). This limitation stems from that the fact that AR models require text-image-to-image triplet data (that is, high-quality annotated data containing reference style images, prompts, and corresponding stylized generated images) in an instruction-following tuning manner to perform style-aligned text-to-image generation, while existing public datasets (Schuhmann et al., 2022) predominantly provide text-image binary data.

Existing approaches for style-aligned text-to-image generation which are predominantly diffusion-based and can be broadly categorized into two paradigms. 1) Optimization-free direct inference methods (Ye et al., 2023; Wang et al., 2024a; Huang et al., 2025), exemplified by IP-Adapter (Ye et al., 2023) which trains decoupled cross-attention layers to integrate style features with text conditions, allowing zero-shot style transfer. 2) Methods requiring parameter tuning or latent inversion (Jeong et al., 2024; Hertz et al., 2024; Ruiz et al., 2023), such as Dreambooth (Ruiz et al., 2023) which performs low-rank (Hu et al., 2022) adaptation (LoRA) for each new reference style, and StyleAligned (Hertz et al., 2024) which extracts the image feature of reference images via DDIM inversion (Mokady et al., 2023), then enforces style alignment through shared attention layers. In contrast, AR models remain underexplored for style-aligned text-to-image generation. Compared to diffusion models (Podell et al., 2023; Rombach et al., 2022) that flexibly blend multimodal features in latent space, AR models (Liu et al., 2024; Team, 2024; Sun et al., 2024) face dual challenges: their heavy dependence on text-image-to-image triplet data and the inefficiency of extraction of style feature via image tokenizer, which collectively hinder their competitiveness in this domain. To overcome these limitations, we present **StyleAR**, the *first* study to enable AR models to perform style-aligned text-to-image generation. We primary focus on two innovative aspects, i.e., data curation and designs of AR models.

On the data aspect, while we can use diffusion-based methods for creating triplet data for instruction-following tuning, they suffer from quality issues due to two main reasons. First, the generated stylized images exhibit low style consistency with the input reference style image, which limiting dataset quality. Second, the generated triplet data serve as ground truth for instruction-following tuning, making the capability frontier of diffusion models an upper bound for AR models, thus challenging the purpose of tailoring AR models for style-aligned text-to-image generation. To address these issues, we propose to drop the reference style image in the data generation process and use only the prompt and the generated stylized image for binary data construction, illustrated as Fig. 2(a). Moreover, we discover that mixing raw images drawn from large-scale text-image dataset (Schuhmann et al., 2022) with stylized images at an optimal 3:1 ratio during training stage can significantly improve our StyleAR's ability to extract richer stylistic features, illustrated as Fig. 2(b)(c). Using these curated stylized image data along with raw image data, we can create a high-quality dataset for AR model tuning.

The remaining part focuses on customizing AR models for style-aligned image generation using binary data. We introduce two key innovations to enhance the AR model to learn from binary data and improve inference performance. **1)** We integrate the image encoder (Radford et al., 2021) with a perceiver (Jaegle et al., 2021; Alayrac et al., 2022) resampler module to convert the input image

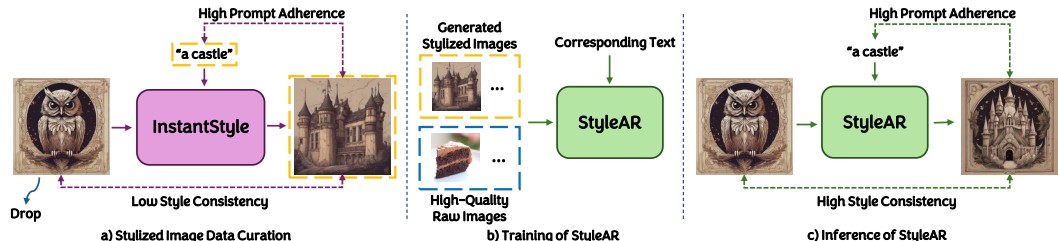

Figure 2: **The pipeline of our method. a)** We first investigate a novel stylized image data curation to form binary data with high prompt adherence and prevent low style consistency. **b)** We use a mixed dataset to enhance rich stylistic features learning. **c)** With the designed data curation and model framework, our method achieve high prompt adherence and style consistency.

to a unified token space that integrates effectively with multimodal tokens. **2)** We introduce a style-enhancing mechanism for tokens using SAM (Kirillov et al., 2023) and Gaussian noise injection to remove irrelevant semantic features from the reference style image.

The key contributions of this work can be summarized as follows:

- We investigate a novel data curation method for crafting binary stylized text-image data from a diffusion counterpart model for training AR models, achieving high quality and avoiding upper bound limit of the base data generation model.

- We propose a training framework for AR models to use binary text-to-image data to perform stylized-aligned text-to-image generation, preventing the difficulty of scaling up text-image-to-image triplet data for instruction-following tuning.

- We also propose the style-enhanced tokens technique, which can effectively solve the problem of content leakage in the stylized-aligned text-to-image generation task and significantly improve prompt adherence and style consistency.

- Extensive quantitative evaluation, qualitative experiments, and user study demonstrate that our StyleAR achieves state-of-the-art performance in both prompt adherence and style consistency, surpassing existing diffusion-based approaches. Moreover, StyleAR integrates additional conditions effectively, such as depth map and other structural control.

## 2 RELATED WORK

### 2.1 MULTIMODAL UNDERSTANDING AND GENERATION UNIFIED MODELS

In recent times, unified models (Team, 2024; Liu et al., 2024; Chern et al., 2024; Wang et al., 2024b; Chen et al., 2025; Zhou et al., 2024; Tang et al., 2024; Fan et al., 2024; Xie et al., 2024; Xiao et al., 2024; Wu et al., 2025; Sun et al., 2025) for multimodal generation and understanding have achieved significant development. These works fall primarily into two categories: autoregressive-based models represented by Chameleon (Team, 2024), and diffusion-based models exemplified by Transfusion (Zhou et al., 2024). In the first category, Chameleon (Team, 2024) pioneered an early fusion approach in which all multimodal inputs are projected onto a shared representational token space from the start, training a transformer to process sequences of multimodal tokens. Building on this foundation, Lumina-mGPT (Liu et al., 2024) enhances training datasets and achieves flexible image generation of varying aspect ratios, and Emu3 (Wang et al., 2024b) adopts a two-stage training framework and introduces the DPO (Rafailov et al., 2023) (Direct Preference Optimization) algorithm to improve visual quality and prompt alignment. Furthermore, Janus-Pro (Chen et al., 2025) decouples visual encoding for image understanding and image generation tasks, employing different image encoders for each objective. HART (Tang et al., 2024) and FLUID (Fan et al., 2024) explore integrating continuous tokens into AR models to improve the quality of image generation. In the second category, Transfusion (Zhou et al., 2024) innovatively combines autoregressive and diffusion training paradigms within a single transformer framework: images are trained via diffusion process, while text follows next-token prediction pattern. Furthermore, Show-o (Xie et al., 2024)

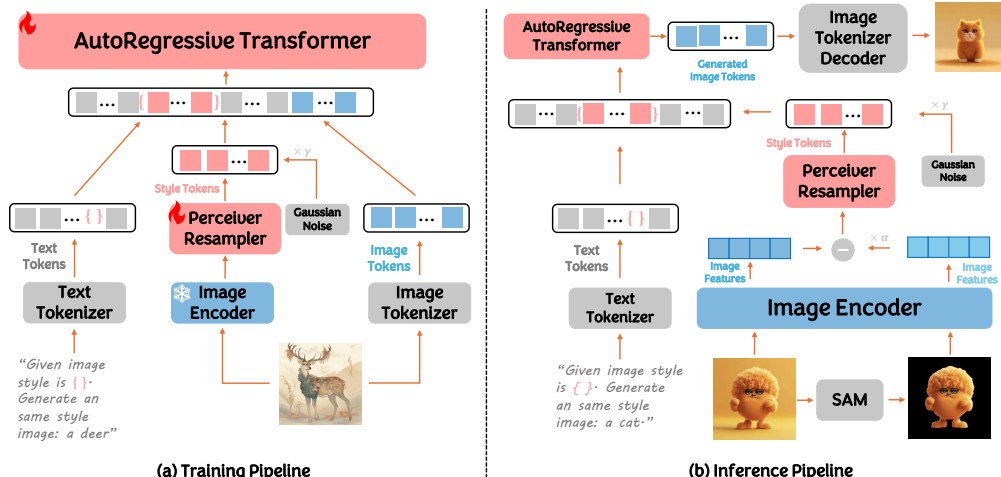

Figure 3: **The framework of our StyleAR.** During training, we utilize a frozen CLIP (Radford et al., 2021) image encoder along with a trainable perceiver (Jaegle et al., 2021; Alayrac et al., 2022) resampler module to efficiently extracted features. Subsequently, style tokens are combined with the injected Gaussian noise and concatenated with multimodal tokens by replacing the placeholder tokens. During inference, we incorporate SAM (Kirillov et al., 2023) to remove irrelevant semantic contents in the reference style image.

and OmniGen (Xiao et al., 2024) expand this approach by scaling the training datasets. In particular, OmniGen (Xiao et al., 2024) incorporates diverse computer vision tasks (e.g., inpainting, deblurring) and subject-driven image generation task into its training data, enabling strong performance across multiple common benchmarks.

## 2.2 STYLE-ALIGNED TEXT-TO-IMAGE GENERATION

Existing approaches for style-aligned text-to-image generation are predominantly diffusion-based and can be broadly categorized into two paradigms. 1) Optimization-free methods (Ye et al., 2023; Wang et al., 2024a; Huang et al., 2025; Hu et al., 2024), exemplified by IP-Adapter (Ye et al., 2023) which trains decoupled cross-attention layers to dynamically integrate style features from pretrained image encoders (Radford et al., 2021) with text conditions, allowing zero-shot style transfer. Similarly, InstantStyle (Wang et al., 2024a) identifies content-specific and style-specific layers in Stable Diffusion XL (Podell et al., 2023) model, proposing a layer-wise decoupling mechanism to suppress content leakage from the reference style images. 2) Methods (Jeong et al., 2024; Hertz et al., 2024; Ruiz et al., 2023) require parameter tuning or latent inversion (Mokady et al., 2023), such as StyleAligned (Hertz et al., 2024) which extracts image features via DDIM inversion (Mokady et al., 2023) and enforces stylization through attention layers. In contrast, AR models remain underexplored for style-aligned text-to-image generation. Compared to diffusion models (Podell et al., 2023; Rombach et al., 2022) that flexibly blend multimodal features in latent space, AR models (Liu et al., 2024; Team, 2024; Sun et al., 2024) face dual challenges: heavy dependence on instruction-following data and inefficiency of style feature extraction in next-token prediction frameworks via image tokenizer, which hinder their competitiveness.

## 3 METHOD

### 3.1 PRELIMINARIES

During the image generation training process of the AR model, for an input image $x \in \mathbb{R}^{H \times W \times 3}$, it is first quantized into $q \in \mathbb{Q}^{h \times w}$ discrete tokens by an image tokenizer (Van Den Oord et al., 2017; Esser et al., 2021; Yu et al., 2021), where $h = H/p_h$, $w = W/p_w$, $p_h$ and $p_w$ are the downsample ratios of the image tokenizer in the vertical and horizontal directions respectively, and

$q^{(i,j)}$ is the indices of the image codebook. Then the image tokens are flattened into a sequence of length $h \times w$ and is concatenated with the text tokens passed into a Transformer-based (Vaswani et al., 2017) autoregressive model for training. During the inference stage, given the text tokens $t$, the autoregressive model can generate image tokens through next-token prediction:

$$\Pi_{t=1}^{h \cdot w} p(q_t | q_{<t}, t). \tag{1}$$

Finally, the generated image tokens are converted to pixel space via a image decoder.

## 3.2 Data Curation

A primary contribution of our method focuses on the data curation part. If we aim at creating triplet data for instruction-following tuning, while we can use InstantStyle (Wang et al., 2024a) to create such data, they suffer from low style consistency and make the capability frontier of diffusion models an upper bound for AR models. In contrast, we drop the reference style image in the data generation process and use only the prompt and the generated stylized image for binary data construction. In this way, we acquire high-quality stylized binary data and prevent low style consistency. Moreover, through practical experiments, we have found that if we rely solely on this stylized dataset for model training, the model's ability to capture image features during the inference stage is unsatisfactory (as shown in Section 4.3), which leads to the style consistency staying low between the generated images and the reference style images. Moreover, considering model nature during pre-training phase of the text-to-image generation task, the training sets it uses mostly consist of raw images that are not stylized. If we only use the stylized dataset for training, the domain gap brings difficulties to the model training process. In view of this, when training our StyleAR, we simultaneously use the raw image dataset and the stylized image dataset in a certain proportion to serve as training dataset.

## 3.3 Framework of StyleAR

**Training with Binary Data.** The framework of our StyleAR are shown in Figure 3. To enable binary data training, we design the model to use the input image in a self-supervised manner, extracting style features and learning to predict the image tokens of the same image. In particular, the input image $I$ is first processed through CLIP (Radford et al., 2021) image encoder $E_I$ to extract image features. The image features are converted to style tokens $s \in \mathbb{R}^{M \times C}$, which fit the unified token space of the AR model by a perceiver (Jaegle et al., 2021; Alayrac et al., 2022) resampler module $R$, where $M = 16$ is the number of style tokens and $C = 4,096$ is the dimension of the unified token space of the AR model. Additionally, to alleviate the content leakage issue (Jeong et al., 2024), we inject Gaussian noise $n$ to style tokens to weaken irrelevant semantic features and enforce the AR model to pay attention to the semantic information from prompts during image generation. The process of image tokens generation is formulated as $\Pi_{t=1}^{h \cdot w} p(q_t | q_{<t}, t, \hat{s})$, where $\hat{s} = s + \gamma \cdot n$, and $\gamma$ is the strength of Gaussian noise injection.

**Style-Enhanced Inference.** We incorporate the SAM (Segment Anything Model) (Kirillov et al., 2023), represented as $S_I$, combining with the Gaussian noise injection mechanism to form the *style-enhanced tokens* technique to further reduce the risk of content leakage and facilitate accurate and reliable inference. Concretely, the input image $I$ and its segmented image $I_S$ are passed through the CLIP (Radford et al., 2021) image encoder to obtain the corresponding image features $F$ and $F_S$. Via feature subtraction $F - F_S$ to exclude semantic information, the result is mapped to the unified token space through the perceiver resampler module. Moreover, to preserve more fine-grained style features, we also introduced a residual path in the unified token space. The style-enhanced tokens $\hat{s}_e$ can be formulated as follows:

$$\hat{s}_e = \alpha \cdot R(F) + (1 - \alpha) \cdot R(F - F_S) + \gamma \cdot n, \tag{2}$$

where $\alpha$ is the residual ratio of the residual path. This inference mechanism significantly improves the stylized image quality, achieving high prompt adherence and high style consistency.

**Post-Training.** Recent numerous research (Ouyang et al., 2022; Rafailov et al., 2023; Shao et al., 2024) have demonstrated the potential of post-training, mainly via reinforcement learning, to enhance the reasoning capabilities of large language models (LLMs) and human preference alignment. In the field of image generation, whether in diffusion models (Black et al., 2023; Fan et al., 2023; Miao et al., 2024) or AR models (Wang et al., 2024b; 2025), post-training is frequently used to

improve prompt alignment and visual quality of generated images (Xu et al., 2023; Xue et al., 2025; Wallace et al., 2024; Fan et al., 2023). In this work, we utilize Direct Preference Optimization (DPO) algorithm (Rafailov et al., 2023) to boost the prompt alignment in the style-aligned text-to-image generation. Specifically, we implement a standard DPO strategy via ranking data creation. For each prompt, we use our StyleAR to generate two images and use the VLM (Yang et al., 2024a;b) to select the image that better aligns with the semantics of the corresponding prompt from these two images. Based on the scoring results, we construct a triplet $(p_i, x_i^{chosen}, x_i^{rejected})$ for DPO training.

## 4 EXPERIMENTS

### 4.1 EXPERIMENTAL DETAILS

**Details of Model and Dataset.** Our StyleAR is implemented based on the FP-SFT@768 version of Lumina-mGPT (Liu et al., 2024). The raw image training dataset is sourced from the open source dataset (opendiffusionai, 2025). For the stylized image training dataset, we collect 80 distinct artistic style images from the open source dataset (Xiao et al., 2024; wikiart, 2022) and generate 200 semantically diverse images per style using InstantStyle (Wang et al., 2024a), resulting in a total training dataset of 16,000 stylized image data. During each epoch, we randomly sample 10% (49,368 images) data from the raw image dataset and mix with the full stylized image dataset to construct the training dataset. The training configuration employs a batch size of 64 with a learning rate of 2e-5 and the rank of LoRA (Hu et al., 2022) parameters employed in the AR model is set to 4.

**Details about Evaluation Metrics.** Following previous work (Hertz et al., 2024; Liu et al., 2023; Sohn et al., 2023; Huang et al., 2025), we use the CLIP-T (Radford et al., 2021) metric to evaluate prompt adherence, which is the cosine similarity between the CLIP text embeddings of the input prompts and the CLIP image embeddings of the corresponding generated images. CLIP-I and DINO (Zhang et al., 2022) metric are used to evaluate style consistency, which is the cosine similarity between the image embeddings of the reference style images and the image embeddings of the corresponding generated images. To robustly measure performance and generalization capabilities of the methods, we collected 20 diverse reference style images and 50 various prompts including human activities, animals, buildings, vehicle, musical instruments, and furniture. For the evaluation suite, we generate four images per style and per prompt, totaling 4000 images.

### 4.2 COMPARISONS

We perform a comprehensive comparison between our proposed StyleAR (AR-based) with existing methods, including InstantStyle (Wang et al., 2024a), IP-Adapter (Ye et al., 2023), StyleAligned (Hertz et al., 2024), StyleCrafter (Liu et al., 2023), StyleShot (Gao et al., 2024), Flux Kontext (Labs et al., 2025) and the state-of-the-art closed source image editing model Nano Banana. With the exception of Nano Banana, all comparative experiments were conducted using the official open source implementations of the baseline methods, with hyperparameter settings strictly adhering to the configurations prescribed in their respective technical documentations.

**Qualitative Comparison.** The quantitative comparison is demonstrated in Figure 4. According to the results, InstantStyle (Wang et al., 2024a) shows superior prompt adherence, achieving a notable semantic alignment of the input prompt and the generated image. However, it shows inferior style consistency between the generated images and the reference style images. The IP-Adapter (Ye et al., 2023) often exhibits a failure in prompt adherence, where the generated images deviate from the input prompts. This artifact originates from content leakage, a phenomenon in which the semantic content of the reference style image (e.g., aircraft) inappropriately propagates into the image generation process. StyleAligned (Hertz et al., 2024) demonstrates unstable generation outcomes and semantic chaos. StyleCrafter (Liu et al., 2023) and StyleShot (Gao et al., 2024) demonstrate satisfactory prompt adherence, but exhibit notable deficiencies in style consistency. While the results of Flux Kontext (Labs et al., 2025) shows unstable style consistency. In contrast, our StyleAR and the Nano Banana demonstrate excellent prompt adherence and accurately captures the overall and detailed features of the reference style.

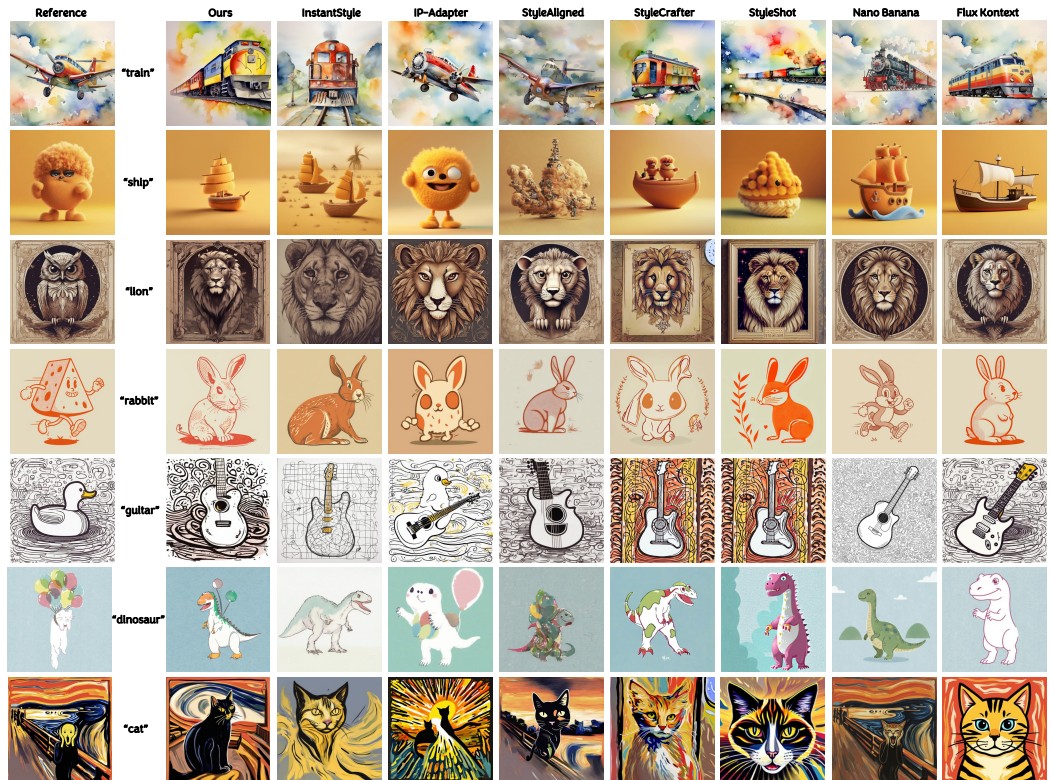

Figure 4: **Qualitative comparison.** We conducted a comprehensive qualitative evaluation by comparing our StyleAR with various existing methods.

Table 1: **Quantitative comparison.** We conduct a comprehensive qualitative evaluation by comparing our StyleAR with various existing methods. The CLIP-T metric reflects the prompt adherence. The CLIP-I and DINO metrics indicate the style consistency. The results consist of the mean and standard deviation of three measurements. Notably, the content leakage issue of IP-Adapter leads to a abnormal increase in CLIP-I and DINO metrics.

|  | StyleAR (Ours) | InstantStyle | StyleAligned | StyleCrafter | IP-Adapter | StyleShot |
|---|---|---|---|---|---|---|
| CLIP-T (↑) | 0.2771±0.0002 | **0.2838±0.0004** | 0.2533±0.0004 | 0.2704±0.0003 | 0.2519±0.0002 | 0.2693±0.0003 |
| CLIP-I (↑) | 0.7319±0.0011 | 0.6642±0.0007 | 0.6771±0.0008 | 0.6824±0.0010 | **0.7726±0.0008**[1] | 0.6718±0.0009 |
| DINO (↑) | 0.6030±0.0016 | 0.4793±0.0022 | 0.5847±0.0019 | 0.5926±0.0015 | **0.6372±0.0017**[1] | 0.5504±0.0015 |

**Quantitative Comparison.** The quantitative comparison results are shown in Table 1. Our StyleAR achieves a superior balance between prompt adherence and style consistency. On the one hand, StyleAR attains the second highest performance in prompt adherence, slightly below InstantStyle (Wang et al., 2024a), while InstantStyle shows poor style consistency. On the other hand, StyleAR ranks second in CLIP-I and DINO metrics, marginally behind IP-Adapter (Ye et al., 2023). However, IP-Adapter (Ye et al., 2023) suffers from poor prompt adherence and severe content leakage (as shown in Figure 4 of qualitative results), which lead to abnormal increases in CLIP-I and DINO metrics. In contrast, our method effectively extracts the style features of the reference style image and generate the target images without any content leakage.

**User Study.** The results of the user study are shown in Figure 5. In terms of prompt adherence and image quality, our method performs on par with InstantStyle (Wang et al., 2024a), both significantly outperforming other methods. Moreover, in terms of style consistency, our method far surpasses all others. In contrast, the InstantStyle (Wang et al., 2024a) method exhibits poor style consistency. It can be seen that our method not only strictly adheres to the input prompts to generate high-quality

---

[1]Abnormal values due to the content leakage issue, which discussed in Section A.3.

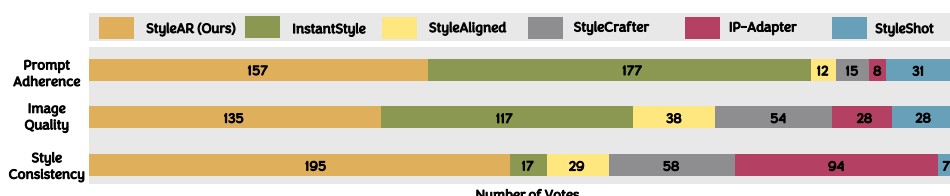

Figure 5: **User study.** We conducted a user study by comparing StyleAR with existing methods.

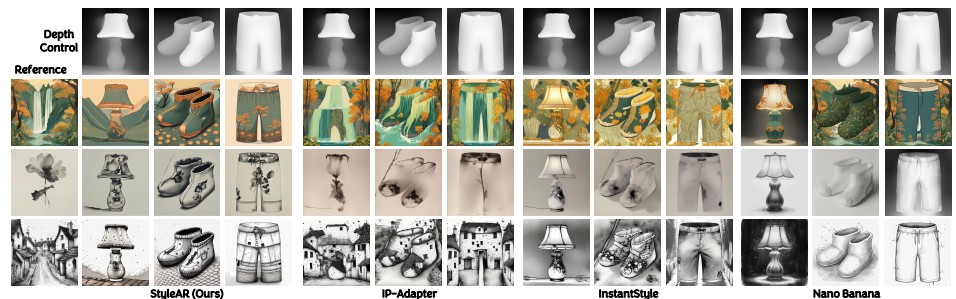

Figure 6: **Qualitative comparison of integration with additional conditions.** We show the comparison results of control generation with Ours and diffusion models.

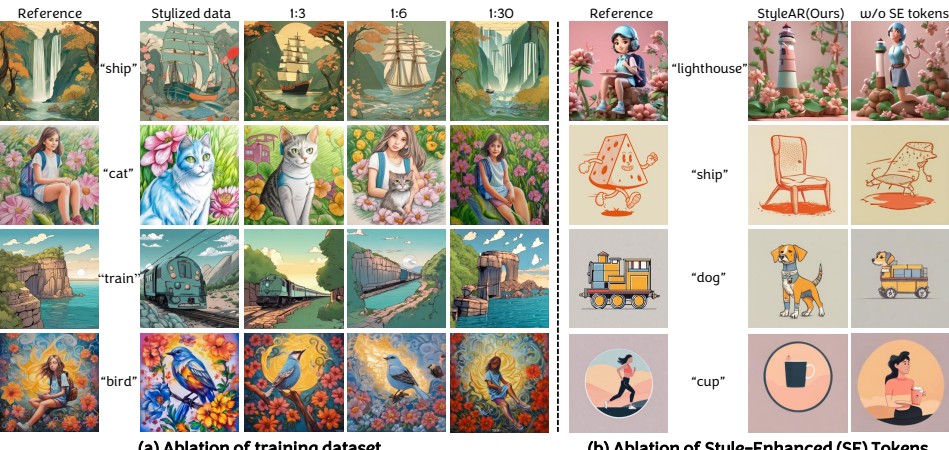

Figure 7: **Ablation study.** We investigate the impact of the composition of training datasets and the style-enhanced tokens technique. The compared training datasets include pure stylized image data, the ratios of stylized image data to raw image data are 1:3, 1:6 and 1:30 respectively.

images but also ensures a high degree of style consistency between the generated images and the reference style images.

**Additional Results.** Freezing the original parameters of the AR model, StyleAR can retain the original generation capabilities of the AR model, such as structural control. Compared with diffusion-based methods like IP-Adapter (Ye et al., 2023) and InstantStyle (Wang et al., 2024a), our StyleAR method performs better on image quality, condition fidelity, and style consistency, shown in Figure 6.

## 4.3 ABLATION STUDY

In this section, we conduct ablation experiments to examine how the training dataset's elements and the design modules impact StyleAR's results.

Table 2: **Quantitative ablation study of composition of training dataset, style-enhanced (SE) tokens and DPO post-training.** The compared variants covers the training dataset study and module designs. The ranking pattern is similar as that in Table 1, indicating the content leakage phenomenon.

| | Stylized data | Ratio 1:3 (Ours) | Ratio 1:6 | Ratio 1:30 | Ours w/o SE tokens | Ours w/o DPO |
|---|---|---|---|---|---|---|
| CLIP-T ($\uparrow$) | **0.2847** | 0.2771 | 0.2568 | 0.2049 | 0.2562 | 0.2754 |
| CLIP-I ($\uparrow$) | 0.6885 | 0.7319 | 0.7338 | **0.7661** | 0.7415 | 0.7014 |
| DINO ($\uparrow$) | 0.5136 | 0.6030 | 0.6551 | **0.7571** | 0.6653 | 0.6086 |

**Composition of training dataset.** We have meticulously designed different compositions of the training datasets to conduct ablation experiments, in order to explore the impact of the training datasets on StyleAR. Specifically, the compared training datasets include pure stylized image data, the ratios of stylized image data to raw image data are 1:3, 1:6 and 1:30 respectively. The qualitative results are shown in (a) of Figure 7 and the quantitative results are shown in Table 2. According to the results, when the training dataset only contains stylized image data, prompt adherence is relatively good, but style consistency is relatively poor. In contrast, when raw images are added, specifically when the ratio of stylized image data to raw image data is 1:3, the style consistency is significantly improved and the prompt adherence also remains at a good level. However, when the ratios are further increased to 1:6 and 1:30, the content leakage occurred and the generated images show "overfitting" to the reference style images. The irrelevant semantic content also appears in the generated images, causing the semantics of the generated images to not follow the semantics of the input prompt. Consequently, we conclude that in stylization tasks, in addition to stylized image data, appropriately adding some raw image data can improve style consistency and will not degrade the prompt adherence.

**Impact of Style-Enhanced Tokens.** To evaluate the effectiveness of our proposed style-enhanced tokens technique, we conduct quantitative and qualitative ablation study to compare our StyleAR and our StyleAR without style-enhanced tokens. The results are shown in (b) of Figure 7 and in the six columns of Table 2. According to the results, when without style-enhanced tokens, the irrelevant semantic features from reference style images show in the generated images, which leads to the generated images not conforming to the semantic control of the input prompt and the situation of chaotic generation occurs. In contrast, prompt adherence and image quality are both improved when style-enhancing mechanism is employed, which enables the style-enhanced tokens to assist the model in filtering out the irrelevant semantic information of reference style images, ensuring that generated images are highly consistent with the input prompt and significantly improving image quality.

**Impact of DPO.** To evaluate the effectiveness of DPO post-training in our StyleAR, we conduct quantitative ablation study to compare our StyleAR and our StyleAR without DPO post-training. The quantitative results are shown in the seventh column of Table 2. Experiments demonstrate that DPO post-training can improve the prompt adherence and enhance the style consistency of StyleAR.

## 5 CONCLUSION AND LIMITATIONS

In this work, we present StyleAR, the first work to use image-text binary data to enable the multimodal autoregressive model to perform style-aligned text-to-image generation which is dominated by diffusion-based methods. Compared with using triplet data in instruction-following tuning of previous AR models, our use of image-text binary data training can easily scale up the size of the training dataset, thereby improving the model's performance. Furthermore, ablation experiments verify the effectiveness of our module designs, including stylized-raw mixed training strategy and style-enhanced tokens technique to enhance style consistency and prompt adherence. However, the current implementation requires depth map extraction for content control rather than direct input of content image to realize style transfer. Future research will focus on leveraging the multimodal input capabilities of autoregressive models to enable simultaneous integration of style reference images and content-specific visual image inputs for further conditional image generation.

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

# A APPENDIX

## A.1 DETAILS OF USER STUDY

To make a more comprehensive comparison, we conduct a user study to evaluate prompt adherence, style consistency, and image quality between different methods. Comparison methods include StyleAR (Ours), InstantStyle (Wang et al., 2024a), StyleAligned (Hertz et al., 2024), StyleCrafter (Liu et al., 2023), IP-Adapter (Ye et al., 2023), StyleShot (Gao et al., 2024). We display 20 pairs of text-image for each participant. Each of them includes a reference style image and the corresponding prompt. We have six randomly generated images of each method for each text-image pair. Each participant is requested to answer three questions for these 20 sets of results: 1) Which generated image's style is the most similar to the reference image's style? 2) Which method produces the highest quality generated images? 3) Which method generates images that best match the input text prompt? The names of all methods have been made anonymous, and we have randomized the order in which the methods appear in each set of responses. Specifically, the study comprised 20 participants, including 10 female and 10 male individuals. Participants encompassed professional artists as well as researchers specializing in image generation—encompassing PhD students and professors—with the majority aged 25–40. This diverse participant composition, spanning both artistic practice and academic research in the domain, is intended to ensure the reliability and comprehensiveness of the user study findings. The number of valid votes we obtained amounted to 1,200.

## A.2 MORE IMPLEMENTATION DETAILS

When fine-tuning StyleAR, the LoRA components are deployed in the QKV and out layers of the self-attention layers, as well as the down-project layer of the MLP layers. Our StyleAR needs to be trained for 10 epochs on a node with eight GPUs. The batch size per GPU is 8, resulting in a total batch size of 64, and it requires 10,000 training steps. The weight decay is set to 0.1, the weight of $z_{loss}$ (Liu et al., 2024) is set to 1e-5 and the gradient clipping is set to 4.

## A.3 FURTHER ANALYSIS OF CONTENT LEAKAGE

In the task of style-aligned text-to-image generation, content leakage has always been a pervasive issue, where not only the stylistic information from the reference style image is transferred to the generated image, but its content information (semantic information) also appears in the generated image. This leads to the content of the generated image being inconsistent with the input prompt or even resulting in chaotic generation. For example, in the experiments of VSP (Jeong et al., 2024), IP-Adapter (Ye et al., 2023) demonstrates higher style consistency metrics (DINO metric in our paper) than VSP (Jeong et al., 2024). However, quantitative results show that the semantic information (content) from the reference image significantly appears in the generated image, causing inconsistencies between the generated image and the input prompt, which is reflected in extremely low text alignment metrics (CLIP-T metric in our paper).

Similar content leakage phenomena are also observed in DEADiff (Qi et al., 2024) and InstantStyle (Wang et al., 2024a). InstantStyle (Wang et al., 2024a), built upon the IP-Adapter (Ye et al., 2023), alleviates content leakage by subtracting the text embedding of the corresponding text from the image embedding of the reference style image. DEADiff (Qi et al., 2024), on the other hand, weakens the content leakage problem by additionally training a Q-Former (Li et al., 2023) to decouple "content" and "style" information of the reference style image.

The experimental phenomena observed in this paper are consistent with previous work (Wang et al., 2024a; Jeong et al., 2024; Qi et al., 2024). For instance, in quantitative comparisons, the IP-Adapter (Ye et al., 2023) method shows very high DINO and CLIP-I metrics, but qualitative results reveal frequent content leakage. Similarly, in our ablation study, quantitative results show that when the ratio of stylized image data to raw image data in the training dataset is 1:30 and SE tokens are not used, DINO and CLIP-I metrics remain very high. However, qualitative result diagrams indicate severe content leakage in both cases.

The current mainstream metrics for measuring style similarity, CLIP-I and DINO, essentially measure the similarity between images, encompassing both stylistic and content aspects. Therefore,

these metrics have limitations in evaluating style similarity for style-aligned text-to-image generation tasks. For example, when content leakage occurs, these metrics produce abnormal results. In the future, it is hoped that with sufficient advancement of Visual Language Models (VLMs). With the emergent capabilities of the base foundation models or aligned capabilities from specially tuned reward models, VLMs can follow fine-grained instructions and align human's intention to decouple image content and style in analysis and accurately determine the stylistic similarity between two images.

## A.4 FURTHER ABLATION ON THE TRAINING DATASET

We conducted further ablation experiments by training our StyleAR framework with only triplet data and comparing it against our method (trained with binary text-image data). The qualitative results are shown in Figure 8, and the quantitative results are presented in Table 3. The results demonstrate that the model trained with triplet data exhibits severely weaker style consistency than our method. This is because the performance ceiling of triplet data training is constrained by InstantStyle (Wang et al., 2024a), which serves as the data generation model, causing it to inherit the inherent drawback of InstantStyle's poor style consistency. In contrast, training with paired data is free from this limitation, leading to excellent style consistency and prompt adherence with our designed binary training for AR and our style-enhanced tokens technique.

Table 3: **Ablation quantitative results on the training dataset.** The compared methods include our method, InstantStyle (Wang et al., 2024a) and our StyleAR trained with triplet data. The CLIP-T metric reflects the prompt adherence. The CLIP-I and DINO metrics indicate the style consistency.

|  | StyleAR (Ours) | InstantStyle | StyleAR Trained with Triplet Data |
|---|---|---|---|
| CLIP-T (↑) | 0.2771 | 0.2838 | **0.2859** |
| CLIP-I (↑) | **0.7319** | 0.6642 | 0.6621 |
| DINO (↑) | **0.6030** | 0.4793 | 0.3876 |

## A.5 BROADER IMPACT

StyleAR empowers users to create distinctive artworks in any stylistic medium with greater controllability. This capability enhances workflow efficiency for professional designers while bringing creative joy to everyday users. However, we recognize the potential risks of misuse, including the creation of sensitive content or copyright-infringing imagery. Therefore, we stress the importance of developing and adhering to ethical guidelines and using this technology responsibly. Additionally, our implementation of watermarking and NSFW classification technologies serves as a proactive safeguard against inappropriate use, ensuring StyleAR remains a force for positive creative expression.

## A.6 ADDITIONAL RESULTS

We present our StyleAR integrate with the additional segmentation map condition applied in Figure 9. Since there is no suitable ControlNet (Zhang et al., 2023) model based on segmentation maps for SDXL (Podell et al., 2023), both InstantStyle (Wang et al., 2024a) and IP-Adapter (Ye et al., 2023) are implemented using the stable diffusion 1.5 version.

## A.7 USE OF LLMS

The only application of LLMs in this work was limited to text polish and linguistic refinement. Specifically, we used LLMs to adjust sentence structures, optimize grammatical accuracy, and enhance the coherence of written expressions. These revisions solely aimed to improve readability while preserving the original meaning of all content. Critically, LLMs were not involved in any core research processes.

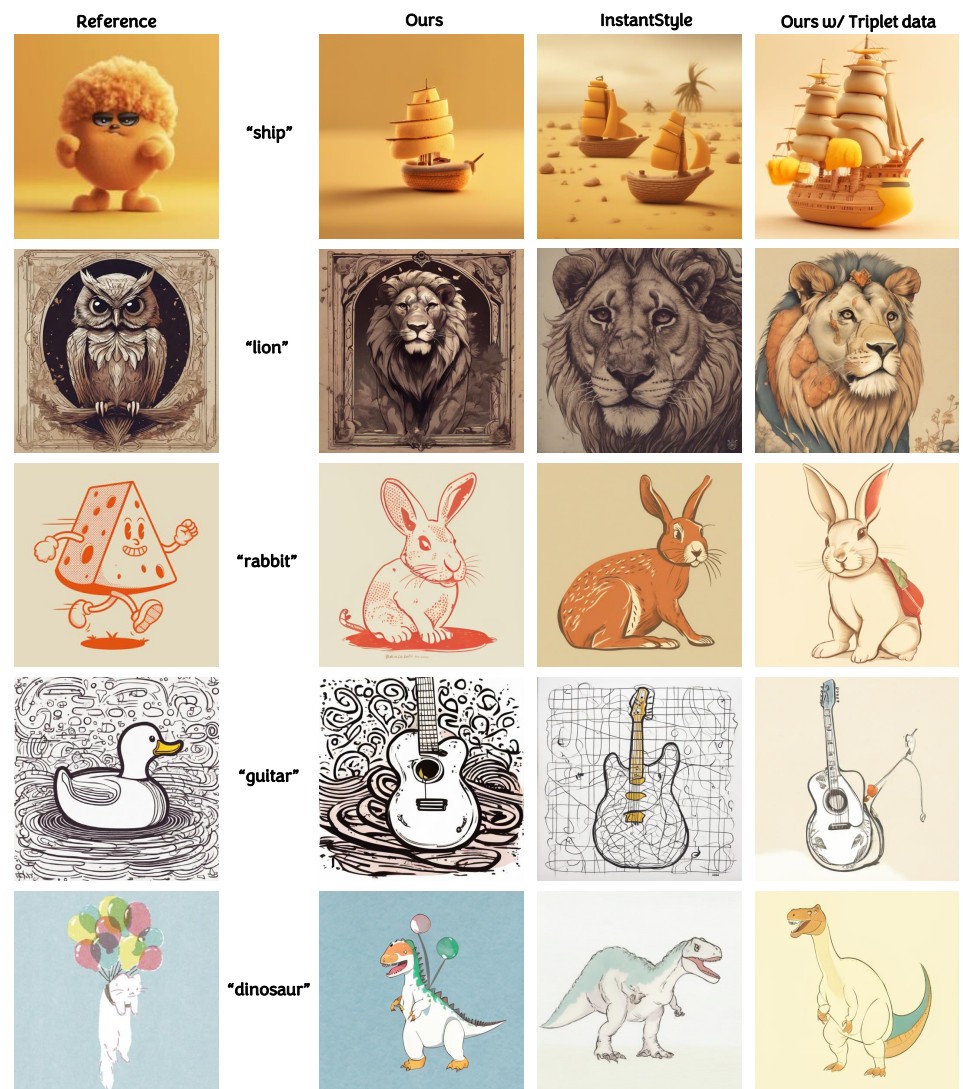

Figure 8: **Ablation qualitative results on the training dataset.** The compared methods include our method, InstantStyle (Wang et al., 2024a) and our StyleAR trained with triplet data.

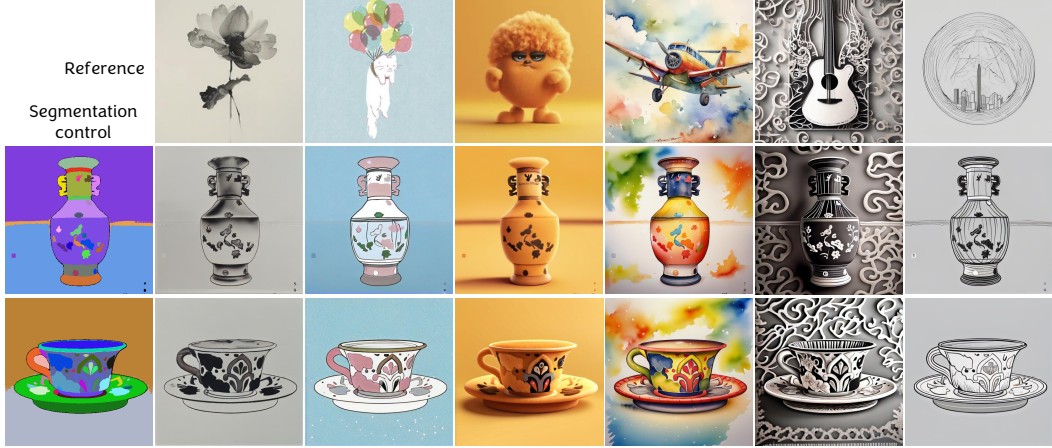

Figure 9: **Qualitative comparison of integration with additional segmentation map condition.**

