# OpenReview forum: "StyleAR: Customizing Multimodal Autoregressive Model for Style-Aligned Text-to-Image Generation"
_ICLR.cc/2026/Conference — Submitted to ICLR 2026_

### Official Review · Reviewer_viPQ · 2025-10-24

**Soundness:** 2
**Presentation:** 2
**Contribution:** 2
**Rating:** 2
**Confidence:** 3

**Summary:**

This paper proposes a new method for enabling stylized text-to-image generation in multimodal auto-regressive (AR) models. Despite the recent progress in unified multimodal generative models, the pre-trained foundation models fail in stylized text-to-image generation, where the model is challenged with controllable generation based on both text prompts and a reference style image. This limitation is due to the lack of high-quality text-image-to-image triplet data. To address this limitation, the authors propose to utilize the text-image binary data, which could be generated from existing diffusion-based stylized generation models. For crafting the style conditions, the authors introduce a Resampler module combined with a CLIP image encoder to extract style features from the training image to augment the text prompts, effectively transforming the task into text-to-image generation.

**Strengths:**

- The presented qualitative samples demonstrate high visual quality, suggesting the efficiency of the proposed method as it accomplishes this text-image-to-image task with only text-to-image data.
- The overall model design is reasonable. To avoid content leakage from the encoded image representation, the authors introduce some Gaussian noise to perturb the generation condition. Such a design helps to improve the robustness of their self-supervised architecture.

**Weaknesses:**

- There is no evidence for the claim ‘the performance of AR models on stylized text-to-image generation is upper bounded by the base data generation model like diffusion models’ in line 129. Specifically, as discussed in lines 228-229 Section 3.2, the low-quality text-image-to-image triplets produced by InstantStyle is the bottleneck for training AR models on this task. To better support this claim, the author shall consider an ablation study on model performance trained on such triplet data compared to that trained on binary data. On the other hand, if the low similarity between style reference image and the stylized output of InstantStyle presents the difficulty, wouldn’t it be simpler to filter the generated data as post-processing, or employ a stronger base model like FLUX? Such a motivation is not sound, especially given the existence of such triplet stylized dataset, or data generation pipeline [1].
- The quantitative metrics for showing the effectiveness of the proposed method is sparse. Currently there are only 3 quantitative metrics adopted. More established quantitative metrics like CSD score [2] would help.
- The novelty of the proposed method is somehow limited. As claimed as the first innovation in line 107, the integration of image encoders and a perceiver resampler is exactly the design choice of IP-Adapter, which is one of the references of this paper.
- The manuscript is a little bit hard to follow. There is substantial space for improvements in the readability. For instance, consider splitting the sentence ‘To facilitate …’ in lines 49-52.

[1] Xing, P., Wang, H., Sun, Y., Wang, Q., Bai, X., Ai, H., Huang, R. and Li, Z., 2024. Csgo: Content-style composition in text-to-image generation. arXiv preprint arXiv:2408.16766.

[2] Somepalli, G., Gupta, A., Gupta, K., Palta, S., Goldblum, M., Geiping, J., Shrivastava, A. and Goldstein, T., 2024. Measuring style similarity in diffusion models. arXiv preprint arXiv:2404.01292.

**Questions:**

The stylized image generation is conditioned on textual representations of prompts augmented by image representations of the style reference. How to balance the generation guidance when there is semantic conflict in these two conditions, for example text prompt ‘a cat’ with a stylized image of a dog?

---

### Official Review · Reviewer_kwRA · 2025-10-29

**Soundness:** 1
**Presentation:** 2
**Contribution:** 1
**Rating:** 2
**Confidence:** 5

**Summary:**

This paper primarily addresses the issue of style transfer in AR models. The authors claim that high-quality AR style transfer models can be trained simply by using binary groups.Freezing CLIP and fine-tuning the resampler are common techniques in diffusion models.

**Strengths:**

The chapter organization of the paper is clear, and the selected images are feasible.

**Weaknesses:**

The introduction in Section 3.2 on training data is not sufficiently addressed. How can binary groups ensure that the prompts cover a wide range of styles? The explanation is very unclear, and it does not convey the efforts made in terms of data.

The training framework seems similar to the approach of diffusion models for style control, lacking novelty.

Many recent state-of-the-art methods were not included in the comparison.

Why is CLIP still used as the style encoder? In fact, there are already some commonly used style encoders, such as CSD, OneIG-StyleEncoder, etc.

Carefully selecting some cases is not very difficult, but I feel that this article does not address the actual challenges.The ablation study is relatively incomplete, as it fails to demonstrate the advantages of the data organization approach in other models.

**Questions:**

see weaknesses

---

### Official Review · Reviewer_S3BH · 2025-10-31

**Soundness:** 3
**Presentation:** 3
**Contribution:** 2
**Rating:** 4
**Confidence:** 4

**Summary:**

This paper introduces StyleAR for style-aligned text-to-image generation using multimodal autoregressive (AR) models. To solve the challenge of acquiring the "text-image-to-image" triplet data required for style transfer task, the author adopts InstantStyle to construct the binary data and combines the raw datasets to joint the model. Besides, the author uses a CLIP encoder with a perceiver resampler to create "style tokens," and a "style-enhanced token" technique designed to prevent content leakage.

**Strengths:**

1. The paper eliminates the need for difficult-to-acquire triplet data (as illustrated in Fig. 3), which can lower the data barrier for style-aligned generation tasks.

2. From the user study and the qualitative results, StyleAR achieves better performance compared to existing approaches.

**Weaknesses:**

1. The idea in this method is a little bit confusing:

a). The paper's core premise is difficult to follow. A central claim is that the method does not require triplet data, yet the authors use InstantStyle to synthesize stylized data for training. This seems contradictory.

b). Furthermore, it is not explained what loss or constraint is used to enforce style consistency during training.

c). The claim that Gaussian noise (n) "weakens irrelevant semantic features" is also questionable, as this noise would likely degrade the essential style information from the reference image simultaneously, undermining the stated goal.

2. The style injection mechanism appears to be adopted from prior work, lacking novelty. More critically, the proposed "Style-Enhanced Inference" method is overly simplistic and suffers from a significant practical limitation: it relies on obtaining a segmented image (mask) of the style reference. This is a non-trivial step and often infeasible for arbitrary style images. The failure of this approach is evident in the qualitative results, particularly in Figure 6 (rows 2 and 4). The 4th row, for instance, exhibits vibrant colors, suggesting the strategy is not robust.

3. Insufficient Experimental Analysis and Unclear Results: The experimental section lacks depth and fails to explain several critical findings:

a). Missing Hyperparameters: The paper does not specify the values for alpha and r (for Style-Enhanced tokens) or describe the methodology for tuning them.

b). Counter-intuitive Ablations: The analysis of training with different ratios of raw vs. stylized images is baffling. Why does increasing the proportion of raw images lead to the model overfitting on style? This is highly counter-intuitive and requires a much clearer explanation. This result raises concerns about potential data leakage: are the styles and prompts used for testing also part of the training set?

c). Persistent Content Leakage: Despite the claims, content leakage is still evident in the generated images (e.g., the balloon on line 343 and the image on line 393). The evaluation of style transfer would be more convincing if the authors tested on more specific or abstract styles (e.g., "fire," "ice") to truly assess the model's ability to separate style from content.

**Questions:**

Please refer to the weakness part.

---

### Official Review · Reviewer_SDzj · 2025-11-01

**Soundness:** 3
**Presentation:** 3
**Contribution:** 2
**Rating:** 4
**Confidence:** 4

**Summary:**

This paper proposes StyleAR, a framework enabling multimodal AR models to perform stylized image generation. A CLIP-based image encoder and perceiver resampler are used to extract style tokens, and a style-enhanced token mechanism is introduced to mitigate content leakage. Experiments show qualitative improvements in style consistency, supported by a user study and ablation analysis.

**Strengths:**

- The qualitative results are strong, showing that the method often produces visually pleasing stylized outputs compared to diffusion-based baselines.
- The proposed style-enhanced token mechanism addresses the content leakage problem that is common in existing style transfer methods.
- The work explores style alignment in autoregressive models, which is a less-studied direction compared to diffusion-based approaches.

**Weaknesses:**

- The methodological novelty appears limited. The overall pipeline mainly involves constructing (text, stylized image) pairs and fine-tuning the AR model, while the style token extraction and integration mechanism resembles an adaptation of existing approaches.
- The proposed model requires training, which makes it computationally more expensive than training-free stylization approaches.
- Quantitative performance lags behind some baselines.

**Questions:**

- In Table 1, it is unclear why NanoBanana and Flux-Kontext are excluded from quantitative metrics.
- The paper would be stronger if it demonstrated that the approach generalizes to multiple AR backbones, not just Lumina-mGPT.
- The method may struggle when style information is localized (e.g., reference image has a stylized object but plain background). Explicit discussion would help clarify limitations and future improvement directions.

---

### Meta-Review · Area_Chair_yhMS · 2026-01-18

**Summary:**

The reviews raise several concerns that collectively outweigh the strengths given the current submission and lack of rebuttal/discussion resolution.

The main decision-driving issues are: (1) novelty/positioning concerns (perceived as incremental or closely aligned with existing conditioning/adapter-style designs), (2) insufficient evidence for key claims and limited ablation support for the proposed components, (3) evaluation gaps (incomplete quantitative validation and/or missing comparisons to relevant recent baselines or metrics for style alignment/content leakage), and (4) generality/practicality limitations (e.g., results shown on a narrow set of settings/backbones and/or requiring training/finetuning where training-free alternatives exist).

Because these are core methodological and empirical questions and there was no rebuttal, reviewers’ negative assessments are unlikely to change, leading to an overall recommendation of reject.

**Reviewer Concerns:**

I did not see a rebuttal, so no concerns were directly

**Reviewer Scores:**

Because there was no rebuttal, I expect little to no upward movement.

---

### Decision · Program_Chairs · 2026-01-26

Reject